# Reliability Characteristics of Metal-Insulator-Semiconductor Capacitors with Low-Dielectric-Constant Materials

**DOI:** 10.3390/molecules28031134

**Published:** 2023-01-23

**Authors:** Yi-Lung Cheng, Wei-Fan Peng, Chi-Jia Huang, Giin-Shan Chen, Jau-Shiung Fang

**Affiliations:** 1Department of Electrical Engineering, National Chi-Nan University, Nan-Tou 54561, Taiwan; 2Department of Materials Science and Engineering, Feng Chia University, Taichung 40724, Taiwan; 3Department of Materials Science and Engineering, National Formosa University, Huwei 63201, Taiwan

**Keywords:** porous film, low-dielectric-constant, reliability, breakdown, time-dependence-dielectric-breakdown

## Abstract

In this study, the reliability characteristics of metal-insulator-semiconductor (MIS) capacitor structures with low-dielectric-constant (low-*k*) materials have been investigated in terms of metal gate area and geometry and thickness of dielectric film effects. Two low-*k* materials, dense and porous low-*k* films, were used. Experimental results indicated that the porous low-*k* films had shorter breakdown times, lower Weibull slope parameters and electric field acceleration factors, and weaker thickness-dependence breakdowns compared to the dense low-*k* films. Additionally, a larger derivation in dielectric breakdown projection model and a single Weilbull plot of the breakdown time distributions from various areas merging was observed. This study also pointed out that the porous low-*k* film in the irregular-shaped metal gate MIS capacitor had a larger dielectric breakdown time than that in the square- and circle-shaped samples, which violates the trend of the sustained electric field. As a result, another breakdown mechanism exists in the irregular-shaped sample, which is required to explore in the future work.

## 1. Introduction

As the feature size of integrated circuits (ICs) is continuously decreased below 0.25 μm, the interconnect resistance/capacitance (RC) delay limits the performance of ICs [1,2]. To overcome this issue, copper (Cu) and low-dielectric constant (low-*k*) materials have been replaced the conventional aluminum (Al) and SiO_2_ materials as a conductor line and an inter-metal dielectric (IMD), respectively [3,4]. The low-*k* materials, which can be achieved by reducing either atomic polarizability or density within the film, provide lower capacitances between the conductors due to their dielectric constant (*k*) less than 4.0. Currently, carbon-doped silicon (C-SiO_2_) film with the incorporation less polarizable Si–H or Si–R termination (where R is an organic moiety such as CH_3_), which is also called organosilicateglass (OSG) film, is widely used in the semiconductor industry. To further reduce the dielectric constant, porous low-*k* materials are produced by introducing nano-pores into the existing OSG low-*k* film. 

As these low-*k* films are integrated in the back-end-of-line Cu interconnects, more challenges arise due to poor mechanical and electrical properties for low-*k* films compared to the traditional SiO_2_ film. On the other hand, with the advance of technology node of ICs, the dimensions of BEOL interconnects are continuously shrinking. However, the supply voltage does not reduce by the same proportion. This leads to a higher electric field being subjected to the low-*k* insulator [5,6]. These effects seriously degrade the dielectric reliability of BEOL interconnects, making low-*k* material reliability to become a bottleneck for new technology node qualification. To overcome this issue, therefore, understanding the reliability characteristics of low-*k* materials is essential and necessary as they are used as IMDs in ICs.

Therefore, the reliability of low-*k* materials is an important topic. This work investigated the reliability characteristics of low-*k* materials using metal-insulator-semiconductor (MIS) capacitor structures. The effects of metal gate area and geometry, thickness of a low-*k* film, and Cu drift were examined. Two kinds of low-*k* materials without and with porosity (called dense and porous low-*k* films) were used. 

## 2. Results and Discussion

### 2.1. Low-k Material Effect

Breakdown of a dielectric film breakdown is a statistical phenomenon and is usually described by the Weibull distribution [7]
(1)$$F(t_{bd})=1-\exp\{-{\left(\frac{t_{bd}}{T_{63.2\%}}\right)}^{\beta }\}$$
where *F* is the cumulative failure probability and is a function of the random variable *t*_bd_ for time-to-breakdown, β is the Weibull slope parameter which governs the breakdown distribution, and T_63.2%_ is the characteristic dielectric breakdown time which represents the time for 63.2% sample failure. Plotting the Weibull function (*W*) can obtain the Weibull plot, given by
(2)$$W=\ln[-\ln(\{1-F(t_{bd})\})]=\beta \ln t_{bd}-\beta \ln T_{63.2\%}$$

This yields a straight line with the slope of β. *T*_63.2%_ can be obtained from *t*_bd_ as *W* equals to 0.

MIS capacitors with dense and porous low-*k* films were stressed at different electric fields. The geometry of the tested MIS capacitors was square-shaped with an area of 9.0 × 10^−4^ cm^2^. During a TDDB test, the leakage current was monitored with stressing time until breakdown in which the leakage current abruptly increased at least three orders of magnitude to more than 10^−2^ A. The stressing time at breakdown is defined as the time-to-breakdown (*t*_bd_). Figure 1a,b plot TDDB results using the Weibull plots for dense and porous low-*k* films, respectively. Both dielectric films exhibited their breakdown times decreasing with increasing electric field, indicating that the electric field is an important factor in controlling breakdown of a dielectric film. Comparing the breakdown times of dense and porous low-*k* films indicates that the dense low-*k* film had a longer breakdown time than the porous film.

Figure 2a,b compares the Weibull parameters (β and *T*_63.2%_ ) for dense and porous low-*k* films films obtained from Figure 1. The β values were in the range of 1.5~2.0, which is consistent with other reports [8,9]. Additionally, β increased and *T*_63.2%_ decreased with increasing the stressing electric field. Moreover, the porous low-*k* films had the smaller β and *T*_63.2%_ compared to the dense low-*k* film in a fixed electric field due to the existing pores in the films. According to electric field analyses within the porous dielectric film, the pores possess a higher field than the other sites [10], triggering and accelerating the occurrence of breakdown.

To precisely predict the lifetime of a dielectric film in a low operation electric field, an accurate TDDB model to describe dielectric reliability is of importance. A widely accepted TDDB model is of the general form [11,12]:(3)$$\ln(TTF)=\alpha (-\gamma E^{m})$$
where *α* is a constant that depends on the properties of a dielectric film, *γ* is a field acceleration factor, *E* is the applied electric field, and *m* = 1 for *E* model and *m* = 1/2 for E^1/2^ model. Based on the E or E^1/2^ models, the field acceleration factors and the predicted lifetimes at 1 MV/cm and 2 MV/cm for the dense and porous low-*k* films are listed in Table 1. Irrespective of the E and E^1/2^ models, the extracted field acceleration factor was larger in the dense low-*k* film than in the porous low-*k* film. Moreover, both low-*k* films had a larger field acceleration factor when the E^1/2^ model was applied. Therefore, the predicted lifetimes in the low field were optimistic for the E^1/2^ model compared to the E model, which is consistent with other reports [13,14]. In order to clarify which model is more correct, the errors (*E*^2^) between the experimental data and fitting model are also listed in Table 1. The error calculation is defined as follows: [13]
(4)$$E_{rror}{}^{2}=\sum_{1}^{n}[\log(T_{63.2\%}^{\mathrm{mod}el})-\log(T_{63.2\%}^{\exp}){]}^{2}$$
where *n* is the total number of stress conditions, *T*_63.2%_ are TTF at 63.2% failure for experimental data and fitting model results. The smaller *E*^2^ value represents the more accurate value for the fitting model. As listed in Table 1, the calculated errors were small, with less than 0.003 for both low-*k* films, indicating that both the E and E^1/2^ models are suitable for low-*k* film (SiCOH) TDDB reliability. Additionally, porous low-*k* films had a slightly larger error value, implying that the pores within the low-*k* film would alter TDDB behavior. In order to further verify the accuracy of TDDB model for low-*k* dielectric films, TDDB tests in lower electric fields (approaching operation electric field) will be performed in a future study.

### 2.2. Metal Gate Area Effect

Area effect is an essential topic for low-*k* film TDDB reliability because the dimensions of the interconnects can vary provided they comply with the design rule. In this part, the square-shaped metal gate MIS capacitors with areas of 0.1–2.5 × 10^−3^ cm^−2^ were used. Before TDDB tests, *C-V* measurements were made to calibrate the metal gate area by using the following Equation (5):(5)$$C=\frac{\varepsilon_{0}kA}{d}$$
where *C* is the capacitance measured at 1 MHz, *ε_0_* is absolute capacitivity in vacuum (8.85 × 10^−12^ F/m), *k* is the dielectric constant of a dielectric film, *A* and *d* represent the metal gate area and film’s thickness, respectively. *C-V* measurement results indicated that the capacitance decreased with increasing area of the designed metal gate. Assuming that the dielectric constant and thickness of a dielectric film remains constant without metal gate area independence, the actual area is therefore determined according to Equation (5). Here, the adopted dielectric constants of dense and porous low-*k* films were 3.02 ± 0.05 and 2.56 ± 0.08, respectively, which were determined from Hg probe measurement. Based on Equation (5) and the measured capacitances and dielectric constant, the areas of metal gate can be calibrated. The areas of metal gate in the fabricated MIS capacitors were higher than those of designed metal gate by 5~10% owing to shadow effect.

The fabricated MIS structures with various metal gate areas were stressed at a fixed electric field in TDDB tests. Figure 3a,b exhibit Weibull plots of TDDB TTFs with various metal gate areas for dense and porous low-*k* films, respectively. For dense and porous low-*k* films, the stressing fields were fixed at 8.8 MV/cm and 7.1 MV/cm, respectively. As shown, TDDB TTFs decreased and the distributions became board with increasing metal gate area for both low-*k* films.

The breakdown events satisfy Poisson random statistics independently of the use of cumulative density distribution because the breakdown sites are randomly distributed over the capacitor area [9]. As the capacitors have different areas, *A*_1_ and *A*_2_, a Weilbull scale and two distributions, *F*_1_ and *F*_2_, are vertically shifted based on the following Equation (6):(6)$$\ln[-\ln(1-F_{1})]-\ln[-\ln(1-F_{2})]=\ln(\frac{A_{1}}{A_{2}})$$

The measured failure times were normalized corresponding to an area of 1.0 × 10^−4^ cm^2^ based on Equation (6), as plotted in Figure 4a,b for dense and porous low-*k* films, respectively. For dense low-*k* films, their TDDB TTFs in various areas can be merged to a single Weilbull distribution, suggesting that the breakdown is intrinsic and the failure site is randomly distributed across the dielectric film. On the other hand, for porous low-*k* films, the transformed distribution deviated from a single Weilbull distribution, especially for a larger area. The obtained TDDB features in the porous low-*k* film are strongly associated with the pores. For the porous low-*k* film in the capacitor with a larger area, the pores are likely to be highly connected, thereby accelerating breakdown. From the merge Weilbull distributions, the β values were determined to be 1.57 and 1.20 for dense and porous low-*k* films, respectively. The determined β value is very useful to forecast the failure times of a dielectric film in the capacitors with various areas according to the following relation:(7)$$\frac{T_{63.2\%,2}}{T_{63.2\%,1}}={\left(\frac{A_{1}}{A_{2}}\right)}^{\frac{1}{\beta }}$$

Figure 5 plots the estimated *T*_63.2%_ values of MIS capacitors with various areas for dense low-*k* and porous low-*k* films. The measured *T*_63.2%_ values at three metal gate areas are also presented for comparison. It is evident to note that the predicted *T*_63.2%_ values are fitted well with the experimental data (within 10% error), demonstrating that Poisson area scaling is valid for the studied dielectric films.

### 2.3. Metal Gate Geometry Effect

TDDB tests were also carried out on the circle- and irregular-shaped MIS capacitors with the porous low-*k* film. The actual metal area was also calibrated by *C-V* measurements. The stressing field in the TDDB test was fixed to be 7.1 MV/cm. To compare TDDB performance for various metal gate geometries, the measured TTFs were correlated based on the metal gate area of 1.60 × 10^−3^ cm^2^ according to Equation (7). Figure 6 compares the *T*_63.2%_ for MIS capacitors with various metal gate geometries. The order of TTF was: Irregular > Circle > Square.

As a constant voltage is applied on the square-shaped MIS capacitors during a TDDB test, it is likely to produce a higher electric field at the edge of metal gate due to the effect of point discharge. As a result, the porous low-*k* films in the square-shaped MIS capacitors sustain a higher electric field, resulting in shorter TTFs. By ANSYS simulation, the sustained electric field within the porous low-*k* film in the MIS capacitors with various metal gate geometries can be obtained, as shown in Figure 7. In the square-shaped MIS capacitor, the maximum sustained electric field is as expected occurred at the edge of metal gate and higher than that in the circle-shaped MIS capacitor, which is responsible for the lower TDDB lifetimes.

On the other hand, the porous low-*k* film in the irregular-shaped MIS capacitors had the highest maximum sustained electric field, possibly due to more edges, but had the longer TTFs than that in the circle- and square-shaped MIS capacitors. As shown in Figure 6, for three irregular-shaped MIS capacitors, the order of TTF was: II > III> I, which is consistent with the trend of maximum sustained electric field. The results illustrate that the sustained electric field within a dielectric film is an important factor in dominating dielectric breakdown. However, irregular-shaped MIS capacitors have an additional mechanism to control low-*k* dielectric’s reliability. In order to explore this mechanism, further experiments will be conducted in the future.

### 2.4. Dielectric Thickness Effect

In this part, the dense and porous low-*k* films with thicknesses from 100 to 550 nm were used to test. The geometry of the tested MIS capacitors was square-shaped with an area of 9.0 × 10^−4^ cm^2^. The breakdown field was determined from *I-V* measurements. Figure 7 plots the dielectric breakdown field (*E*_B_) versus the physical thickness for dense and porous low-*k* films. The breakdown field increased with decreasing the thickness for both low-*k* films. Under electric stress, charge trapping and de-trapping simultaneously occur in a dielectric film. As the trapped charges reach the critical value to form a conductive path within a dielectric film, breakdown occurs. As a result, as the thickness of a dielectric film decreases, de-trapping is becoming active, leading to an increase of breakdown field. As the thickness of a dielectric film is further scaling-down, the de-trapping rate would exceed the trapping rate, making a dielectric breakdown not occur [14]. Accordingly, a dielectric film has a critical thickness (*d*_c_) at which the de-trapping rate of charges equals the trapping rate. A dielectric film whose thickness is less than this critical thickness does not exhibit dielectric breakdown. Since the dielectric breakdown strength is inversely proportional to the physical thickness, the thickness-dependent dielectric strength can be expressed as Equation (8):(8)$$E_{B}\;=A{\left(\;d-d_{c}\;\right)}^{-n}$$
where *A* is the constant, *d* is dielectric physical thickness, and *n* represents the power-law constant. The fitted *n* values from the slope of the line in Figure 8 were 0.324 and 0.251 for the dense and porous low-*k* films, respectively. These obtained values were in the reported range of 0.2~1.0 [14]. The dense low-*k* films had a higher *n* value than the porous low-*k* films, indicating that the breakdown strength of the dense low-*k* films is strongly affected by their thicknesses. For the porous low-*k* film, its breakdown strength is degraded by the existing pores. Hence, the impact of the thickness on the breakdown strength is not as strong as that of the dense low-*k* film. The obtained critical thicknesses of the dense and porous low-*k* films, which were calculated using the least-squares method, were 12.90 nm and 7.86 nm, respectively. For the porous low-*k* films, the existing pores act as conducting sites, thereby reducing the critical number of electronic charges that are required to form a conducting path between the cathode and the anode and leading to a lower critical thickness.

The breakdown times of the dense and porous low-*k* films with thicknesses of 100~560 nm were measured. The E-model was used to predict the failure time in a low operating electric field of 1.0 MV/cm. In order to obtain γ values, three fields were stressed for each condition. The TDDB results are shown in Figure 9.

The dense low-*k* dielectric films had larger γ values as than porous low-*k* samples. Additionally, the obtained γ values increased as the physical thickness decreased. A larger γ value can result in a longer dielectric failure time in a lower electric field. Based on the TDDB results, the dielectric breakdown times of the dense and porous low-*k* films in an electric field of 1.0 MV/cm, predicted using Equation (3), as a function of thickness are plotted in Figure 10. As expected, the predicted dielectric breakdown times of the low-*k* films increased with decreasing the film thickness. Additionally, the dense low-*k* films had longer dielectric breakdown times in an electric field of 1.0 MV/cm than did the porous low-*k* dielectric films. This improvement ratio of the dielectric breakdown time between dense and porous low-*k* dielectric films increased as the dielectric film became thinner. Since a thinner low-*k* film exhibited a pronounced de-trapping effect, the thickness-dependent dielectric breakdown time in an electric field of 1.0 MV/cm satisfies the dielectric strength equation, as given by Equation (9):(9)$$t_{bd}\;=f{\left(d-d_{c}\right)}^{-m}$$
where $\;t_{bd}$ is the dielectric breakdown time in an electric field of 1.0 MV/cm, *f* is the constant, and *m* represents the power-law constant. The *d_c_* values of the dense and porous low-*k* dielectric films, obtained from Equation (7), were 12.90 nm and 7.86 nm, respectively. Based on the predicted dielectric breakdown times and Equation (9), the extracted *m* values of the dense and porous low-*k* dielectric films were 2.868 and 1.067, respectively.

## 3. Experiments

Both dense and porous low-*k* film were deposited on 300 mm *p*-type Si wafers using applied material plasma-enhanced chemical-vapor-deposition reactor with a radio frequency (rf) of 13.56 MHz at a temperature of 300 °C. Both low-*k* films, which use diethoxymethylsilane (DEMS) and oxygen (O_2_) as the deposition matrix precursors, were SiCOH materials. To produce porous low-*k* films, alpha-terpinene (ATRP) was also introduced into the reactor as a porogen precursor. After deposition, UV thermal-assisted curing with 200~450 nm wavelength was performed to remove the organic porogen to form the pores in the film. Details about the deposition conditions of dense and porous low-*k* films can be found elsewhere [15,16]. The basic properties of the dense and porous low-*k* films are listed in Table 2.

The thickness and refractive index of both low-*k* films was analyzed on an optical-probe system with an ellipsometer (Film Tek^TM^ 3000SE). The pore size and porosity were determined from the isotherm of ethanol adsorption and desorption using ellipsometric porosimetry (Semilab, Mode PS-1100). The dielectric constants of films were calculated from capacitances of the MIS structure at the voltage of 0.1 V and the frequency of 1 MHz using a semiconductor parameter analyzer (HP4280A).

After the deposition of dielectric films, a metal electrode (Al) with a thickness of ~10 nm was deposited on the top surface of the films through shadow mask using a thermal evaporation method. The patterns used in the shadow mask are listed in Table 3, including the area and the geometry. The fabricated metal-insulator-silicon (MIS) capacitors were used to measure the leakage current voltage (*I-V*), capacitance voltage (*C-V*) characteristics, and dielectric-dependence dielectric breakdown (TDDB) failure time. The measurement tools used were a semiconductor parameter analyzer (HP4280A) and an electrometer (Keithley, 6517A). All measurements were performed at room temperature (25 °C).

## 4. Conclusions

The reliability characteristics of dense and porous low-*k* films have been investigated and compared in this study. The effects of metal gate area and geometry and low-*k* film thickness are studied. Compared to the dense low-*k* film, the porous low-*k* film has a shorter TDDB lifetime, a lower β value and electric field acceleration factor, and a weaker thickness-dependence breakdown. Additionally, it has a larger deviation in the breakdown projection model (E and E^1/2^ models) and single Weilbull plots of the breakdown time distributions from various areas merge. Finally, the dielectric breakdown time is also affected by the geometry of the metal gate. The porous low-*k* film in the irregular-shaped metal gate MIS capacitor had a larger dielectric breakdown time than that in the square- and circle-shaped samples, which violates the trend of the sustained electric field. A clear breakdown mechanism for the irregular-shaped sample has not been made clear. Hence, further experiments will be carried out in the future.

## Figures and Tables

**Figure 1 molecules-28-01134-f001:**
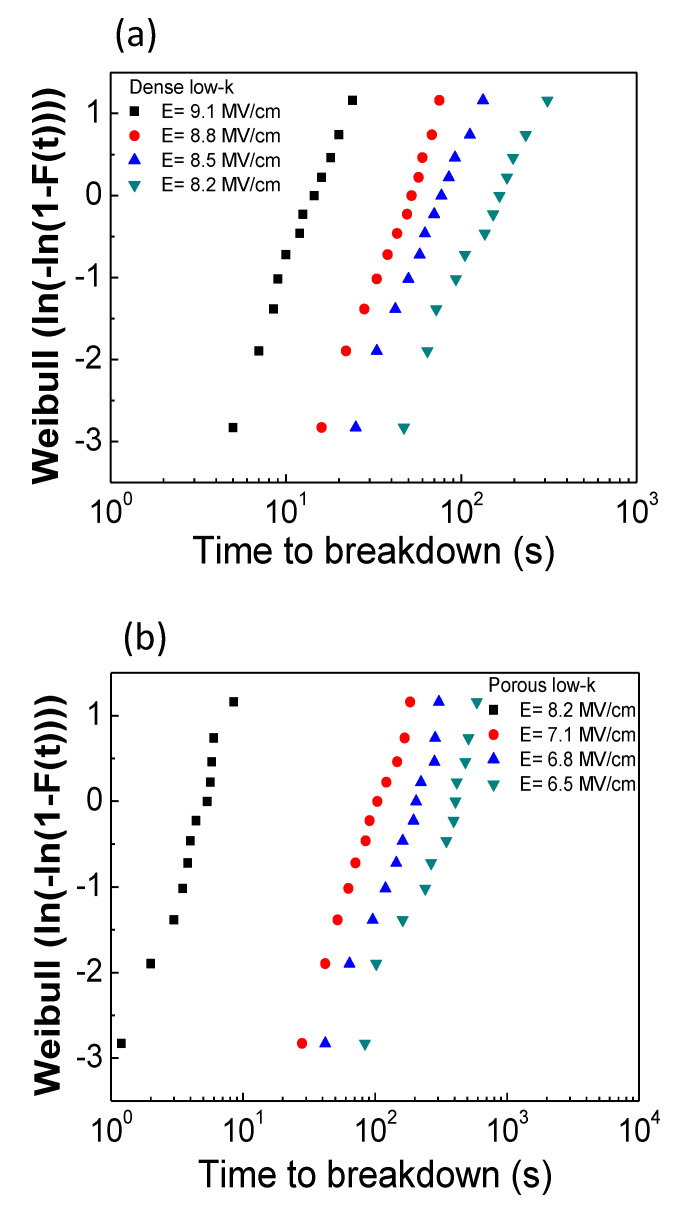
Weibull plots of TDDB results: (**a**) dense low-*k*; (**b**) porous low-*k*.

**Figure 2 molecules-28-01134-f002:**
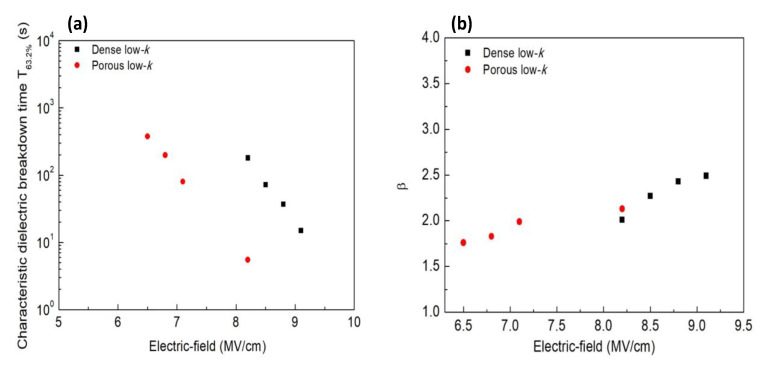
Weibull parameters for dense and porous low-*k* films: (**a**) T_63.2%_; (**b**) β.

**Figure 3 molecules-28-01134-f003:**
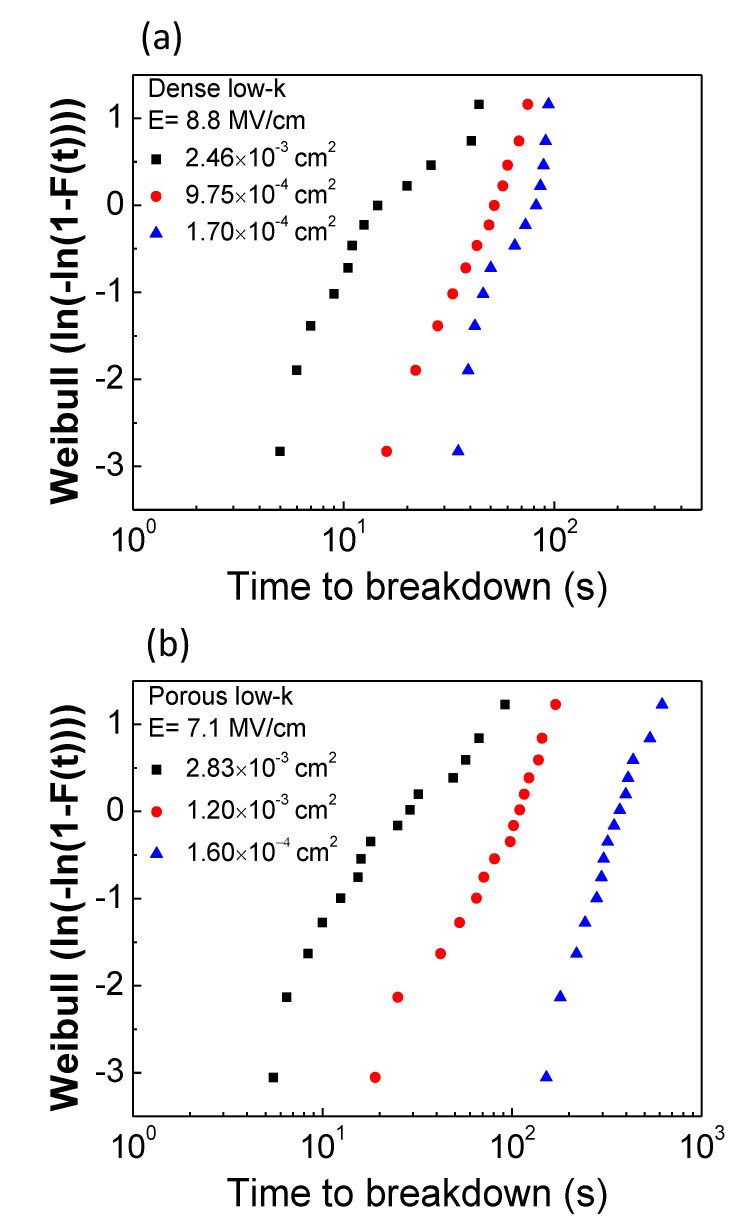
Weibull plots of TDDB results for various gate areas: (**a**) dense low-*k*; (**b**) porous low-*k*.

**Figure 4 molecules-28-01134-f004:**
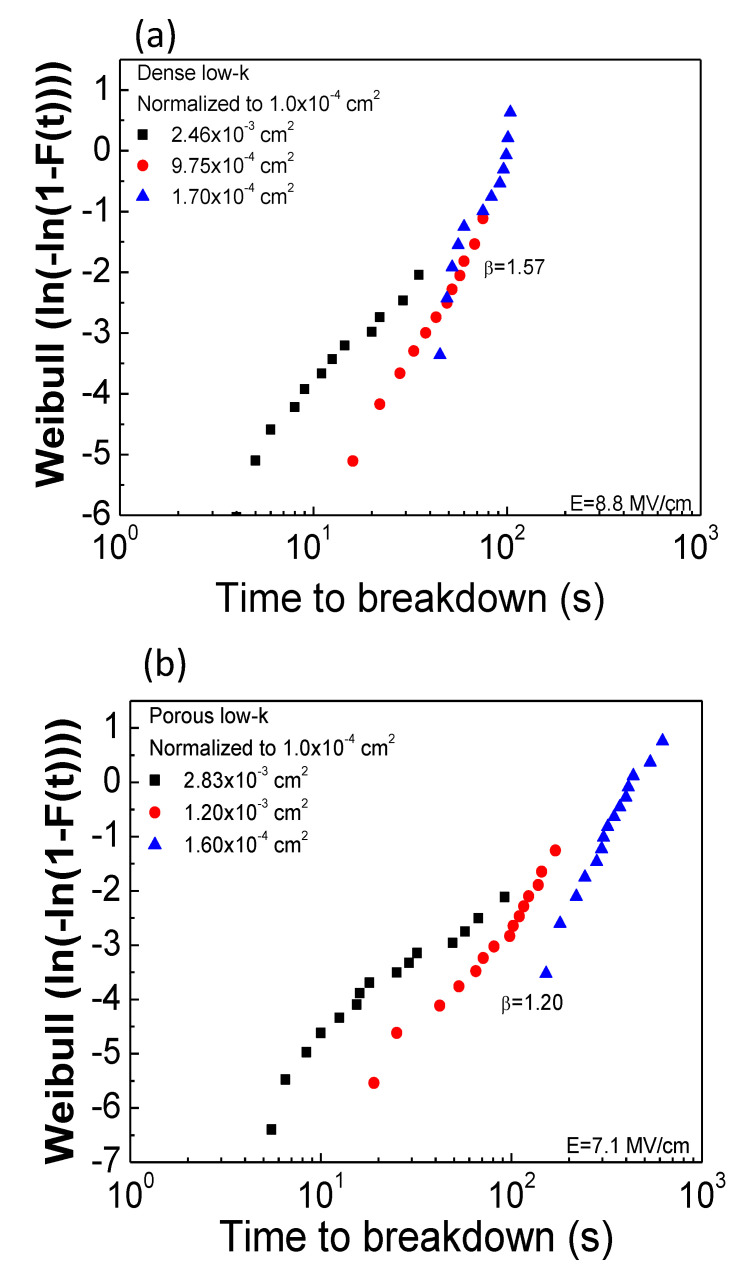
Normalized Weibull plots of TDDB results: (**a**) dense low-*k*; (**b**) porous low-*k*.

**Figure 5 molecules-28-01134-f005:**
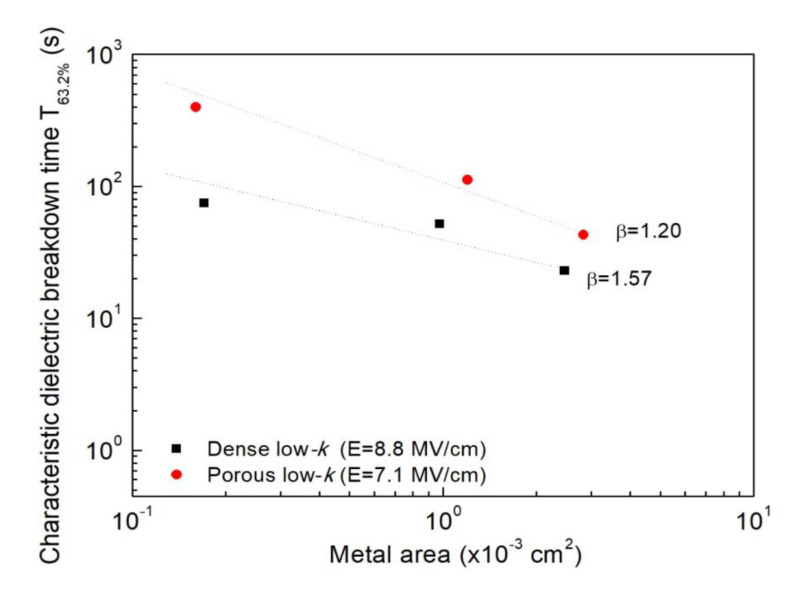
The predicted TDDB lifetime at various gate areas for dense and porous low-*k* films.

**Figure 6 molecules-28-01134-f006:**
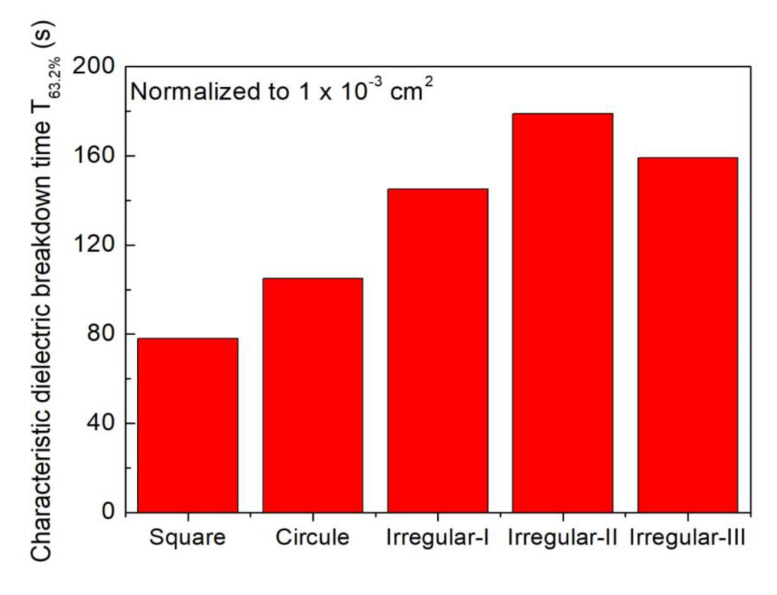
Comparison of TDDB lifetimes for porous low-*k* film in various gate geometries.

**Figure 7 molecules-28-01134-f007:**
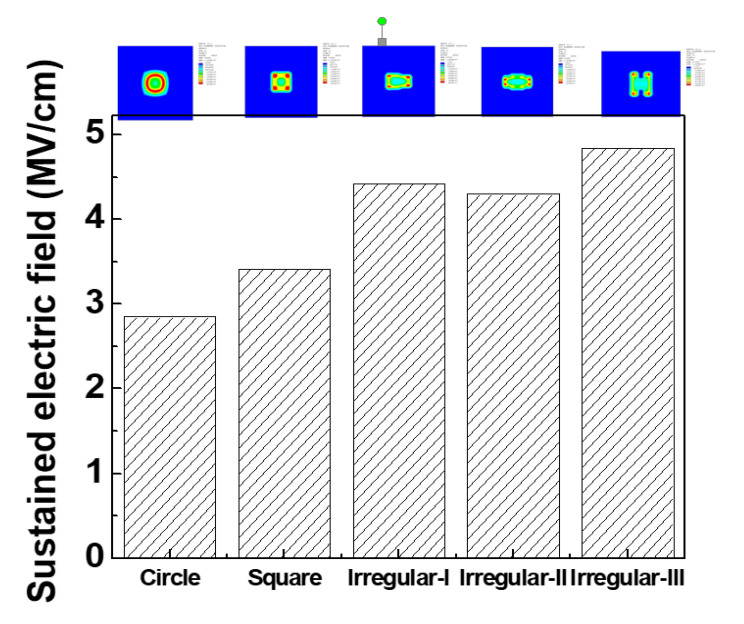
Maximum electric field within porous low-*k* film for various gate geometries.

**Figure 8 molecules-28-01134-f008:**
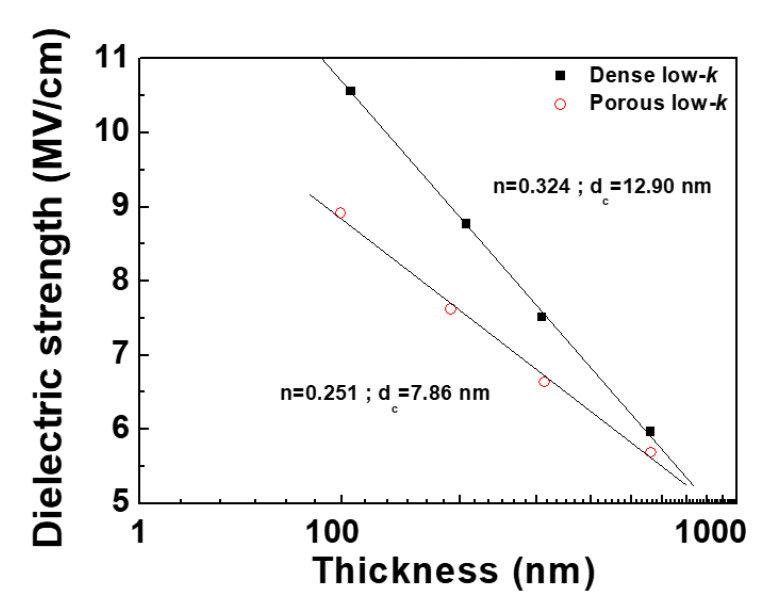
Thickness-dependent dielectric breakdown strength for dense and porous low-*k* films.

**Figure 9 molecules-28-01134-f009:**
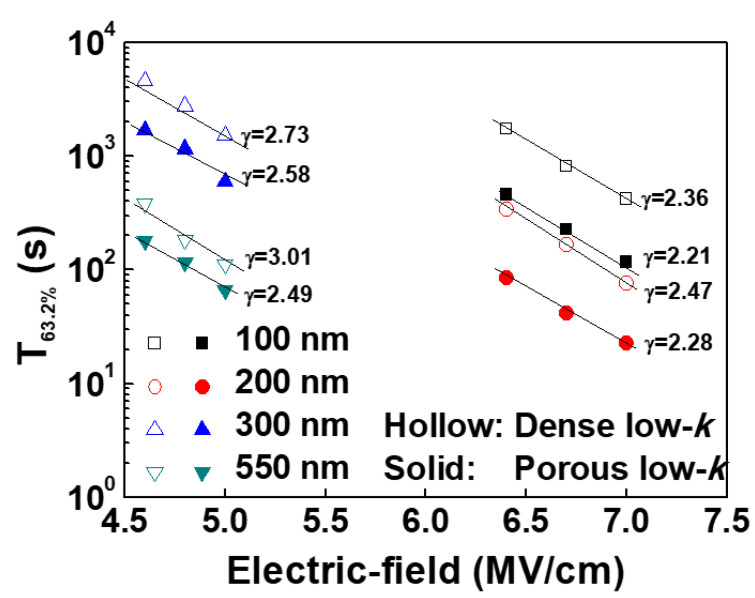
Characteristic dielectric breakdown time for dense and porous dielectric low-*k* films with various thicknesses as a function of stressing electric field.

**Figure 10 molecules-28-01134-f010:**
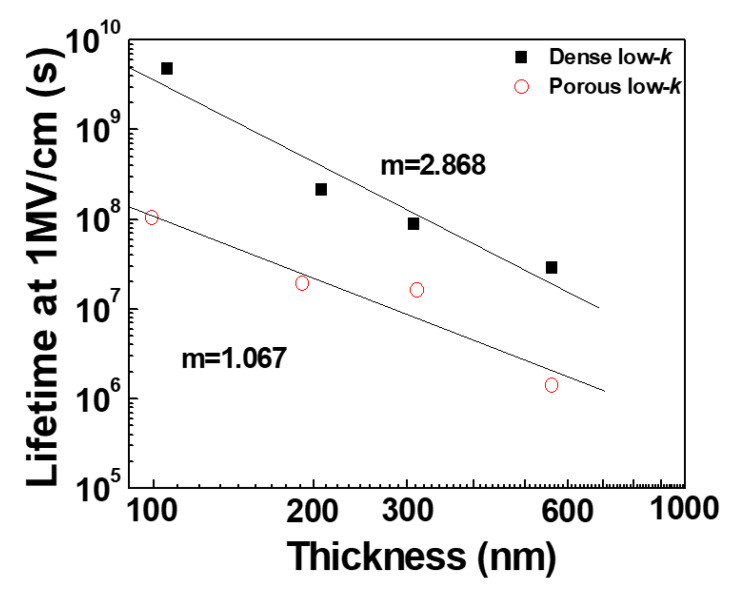
Predicted dielectric failure time at 1.0 MV/cm for dense and porous low-*k* films as a function of thickness.

**Table 1 molecules-28-01134-t001:** *E* model or *E*^1/2^ model parameters for dense and porous low-*k* films.

Dielectric	Dense Low-k		Porous Low-k	
Model	*E*	*E* ^1/2^	*E*	*E* ^1/2^
γ	2.70	15.9	2.51	13.61
Error (E_rror_^2^)	0.00229	0.00225	0.00240	0.00253
Lifetime at 1 MV/cm (s)	5.12 × 10^10^	1.36 × 10^15^	3.70 × 10^8^	5.82 × 10^11^
Lifetime at 2 MV/cm (s)	3.44 × 10^9^	2.01 × 10^12^	3.01 × 10^7^	2.19 × 10^9^

**Table 2 molecules-28-01134-t002:** Film properties of dense and porous low-*k* films in this study.

Sample	Precursor	UV Curing	Dielectric Constant (*k*)	Leakage Current Density at 1 MV/cm (×10^−12^)	Breakdown Electric Filed at 25 °C (MV/cm)	Porosity (%)	Pore Size (nm)
Dense low-*k*	DEMS + O_2_	No	3.02 ± 0.05	5.66 ± 0.7	9.4 ± 0.5	N/D	N/D
Porous low-*k*	DEMS + O_2_ + ATRP	Yes	2.56 ± 0.08	2.69 ± 0.33	8.2 ± 0.4	15.0 ± 0.5	1.35 ± 0.14

N/D = not detected.

**Table 3 molecules-28-01134-t003:** Designed geometry and area of metal gate in MIS capacitor.

Mask Geometry	Square	Circle	Irregular
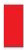	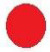	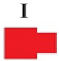	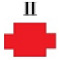	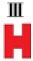
**Designed mask area (×10^−3^cm^2^)**	**0.10~2.50**	**0.90~2.46**	**1.62**	**1.62**	**1.60**

## Data Availability

Not applicable.

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
