# Peer review of "Reliability Characteristics of Metal-Insulator-Semiconductor Capacitors with Low-Dielectric-Constant Materials"

_molecules, 2023, doi:10.3390/molecules28031134_

Round 1

Reviewer 1 Report

In this paper the authors study the reliability characteristics of metal-insulator-semiconductor capacitor structures with low dielectric constant materials, focusing on the effect of metal gate area geometry and thickness of dielectric film. The section concerning the results is very extensive but the introduction and most important the experiments are briefly presented. 

My comments to the authors are the following:

1. As I have already mentioned, I believe that the experimental setup and procedure should be described in more detail. A scheme of the experimental setup used for the measurements will be helpful. The leakage current is probably used for the determination of  breakdown of dielectric film but the authors should describe in more clearly the conditions concerning the breakdown event. 

2. In figure 7,  the distribution of the electric field for the different geometries of metal gates in the top of the figure is not visible at all. It should be fixed as it is very important for the interpretation of the experimental results.

Author Response

  1. As I have already mentioned, I believe that the experimental setup and procedure should be described in more detail. A scheme of the experimental setup used for the measurements will be helpful. The leakage current is probably used for the determination of  breakdown of dielectric film but the authors should describe in more clearly the conditions concerning the breakdown event. 

[Reply] Exp. details had been added in the revised manuscript (Yellow marked). In the result part, we had defined breakdown (the leakage current abruptly increased at least three orders of magnitude to more than 10-2 A.)  

  1. In figure 7,  the distribution of the electric field for the different geometries of metal gates in the top of the figure is not visible at all. It should be fixed as it is very important for the interpretation of the experimental results.

[Reply]  We had shown the max. electric field for the different geometries of metal gates in Fig. 7. The purpose of the distribution of the electric field for the different geometries of metal gates in the top of the figure is to display the location of max. electric field. These plots had been revised to make clear.

Reviewer 2 Report

Cheng et al. demonstrated a very interesting and important work. The aim of this study was to investigate the reliability characteristics of low-k materials using metal-insulator-semiconductor (MIS) capacitor structures. Although further research was needed to determine whether another breakdown mechanism occurred in the irregular-shaped sample, low-k film thickness, metal gate area and geometry, and Cu drift were examined. In addition, there were two kinds of low-k materials used, namely dense and porous films. This work meets the demand of the readership, but some revisions are still needed.

1.      What method is used to measure the pore size of the material? How uniform and repeatable are the pores?

2.      Does the experiment explore the effects of material porosity variation on material properties? As an example, you can change the ATRP ratio.

3.      To investigate the effect of metal gate areas on material properties, dense and porous materials with similar voltage influences are selected, such as 8.2V dense and 7.1V porous materials. Can a more detailed comparison be made between porous and dense porous materials based on the metal gate areas?

4.      There are omissions and errors in the references part.

Author Response

  1. What method is used to measure the pore size of the material? How uniform and repeatable are the pores?

[Reply] We added the measure method and error bar for the pore size.

  1. Does the experiment explore the effects of material porosity variation on material properties? As an example, you can change the ATRP ratio.

[Reply] We had changed APTMS/DEMOS ratio to explore the effects of porosity variation. The result had been published in " Effects of Precursor Flow Rates on Characteristics of Low-k SiOC(H) Film Deposited by Plasma-Enhanced Chemical Vapor Deposition", Electrochemical Society Transactions. 72(2), p.253-268 (2016). In this study, we fixed the pore size for the porous low-k films.

  1. To investigate the effect of metal gate areas on material properties, dense and porous materials with similar voltage influences are selected, such as 8.2V dense and 7.1V porous materials. Can a more detailed comparison be made between porous and dense porous materials based on the metal gate areas?

[Reply] In this study, the metal gate area effects between porous and dense low-k films had been compared in Fig. 6. For the metal gate geometry effect, we only performed on porous low-k films.  

  1. There are omissions and errors in the references part.

[Reply] References part had been corrected and revised.

Round 2

Reviewer 1 Report

The authors have made sufficient efforts in order to comply with the comments of the reviewers. The experimental procedure is described in more detail and the conditions of breakdown event are more  clearly defined. Additionaly the plots are revised in order to be more clear and better visible. I think that the article is now properly prepared for publication.

Author Response

The authors have made sufficient efforts in order to comply with the comments of the reviewers. The experimental procedure is described in more detail and the conditions of breakdown event are more  clearly defined. Additionaly the plots are revised in order to be more clear and better visible. I think that the article is now properly prepared for publication.

[ Reply] Thanks for reviewer's positive comment!